# Cost-Effective Detection of SNPs and Structural Variations in Full-Length Genes of Wheat and Sunflower Using Multiplex PCR and Rapid Nanopore Kit

**DOI:** 10.3390/biology14020138

**Published:** 2025-01-29

**Authors:** Ekaterina Polkhovskaya, Evgeniy Moskalev, Pavel Merkulov, Ksenia Dudnikova, Maxim Dudnikov, Ivan Gruzdev, Yakov Demurin, Alexander Soloviev, Ilya Kirov

**Affiliations:** 1All-Russia Research Institute of Agricultural Biotechnology, Timiryazevskaya Str. 42, 127550 Moscow, Russia; eynzeynkreyn@gmail.com (E.P.); badsaxson@mail.ru (E.M.); paulmerkulov97@gmail.com (P.M.); saenkok1997@yandex.ru (K.D.); max.dudnikov.07@gmail.com (M.D.); gruzdev82mtz@mail.ru (I.G.); a.soloviev70@gmail.com (A.S.); 2Pustovoit All-Russia Research Institute of Oilseed Crops, Filatova St. 17, 350038 Krasnodar, Russia; genetic@vniimk.ru; 3All-Russia Center for Plant Quarantine, 140150 Ramenski, Russia

**Keywords:** nanopore sequencing, amplicons, wheat, sunflower, SNPs, InDels

## Abstract

Modern plant breeding relies heavily on harnessing the genetic diversity present in crops, which requires the identification of multiple single-nucleotide polymorphism variants across entire genes of interest. The traditional methods for detecting these variants can be time-consuming, expensive, and often inaccessible for smaller breeding companies and laboratories. In this study, we demonstrate a rapid and cost-effective approach for detecting single-nucleotide polymorphisms and structural variants in full-length target genes by integrating multiplex PCR, a Rapid Barcoding procedure, and nanopore sequencing. We applied this method to analyze genetic variation in four sunflower genes (Ahasl1, Ahasl2, Ahasl3, and FAD2) across 40 genotypes, as well as three wheat genes (Ppo, Wx, and Lox) across 30 genotypes. Our findings provide a comprehensive overview of the genetic variant distribution along these genes, highlighting significant gene diversity within the germplasm collections. This streamlined approach not only enhances efficiency but also reduces costs and labor requirements, making rapid sequencing and genotyping more accessible for plant breeders. By facilitating quicker access to essential genetic information, this method has the potential to accelerate the breeding process and improve crop development outcomes.

## 1. Introduction

The rapid identification of allele variants of target genes in plant collections is essential for accelerating the plant breeding process using marker-assisted selection (MAS). Allele-specific DNA markers are typically used in MAS to detect certain polymorphic regions of a gene. This strategy is rapid and cost-effective owing to the availability of modern detection systems (for example, KASP, ASPCR, and PCR-SSCP) that allow for genotyping hundreds of individual plants [1,2,3,4,5]. These high-throughput genotype systems have been widely used in plant breeding for various crop species [6,7,8,9,10,11,12,13,14,15]. KASP and other systems for high-throughput genotyping, for example, allele-specific PCR (ASPCR) and single-strand conformation polymorphism (SSCP), allow for the discrimination of polymorphisms with a high degree of specificity and sensitivity [4].

The low cost and high throughput of these genotyping approaches make them ‘a way to choose’ when the DNA variant sites linked to certain traits are known and are not scattered along the gene. However, for more diverse germplasm sources involved in breeding, more DNA variant sites in different gene parts have been discovered. For example, glutenin genes have multiple single-nucleotide polymorphism (SNP) variants that are distributed over the entire gene and can differ between distinct Glu alleles [16,17]. Genes that improve the quality of wheat flour, such as lipoxygenase (*Lox*), polyphenol oxidase 1 (*Ppo*), and the granule-bound starch synthase (*Wx*), are of particular interest for allele identification and marker development. Identifying any structural variations in genes associated with baking quality is crucial for the breeding process. Another example of detrimental genes for important traits is the family of *AHASL* genes, which are common targets of herbicides, and crop resistance to different herbicides is associated with distinct SNPs in different gene parts [18,19,20]. PCR-based methods have shown a C-to-T mutation in codon 205 of the *Ahasl1-1* gene, offering moderate resistance to imazamox (IMI) [21,22]. An analysis of *Ahasl1-2* revealed a C-to-T mutation in codon 197, rendering a substantial level of SU (sulfonyl-urea) tolerance [21]. *Ahasl1-3* involves a G-to-A mutation in codon 122, which confers strong resistance to IMI [23]. *Ahasl1-4* contains a G-T mutation in codon 574, granting broad resistance to herbicides targeting *AHAS* in four different families [24]. Nevertheless, herbicide resistance is not restricted to only four alleles [25].

The existence of multiple polymorphic sites in the gene sequence and different variants makes the design of marker systems more complex and expensive [26]. Therefore, another approach based on target gene sequencing (TGS) has been used. Different TGS methods have been developed over the last few years, but they are still time-consuming and require a certain level of molecular biology skills. The TGS methods can be divided into two groups: target amplicon sequencing (TAS) and target DNA ‘fishing’ and sequencing (TAFS). While the former approaches are based on PCR amplification, the latter (e.g., nCATS [27] and hybridization-based capture sequencing [28]) can be used without an amplification step. The biggest disadvantage of TAFS methods is the requirement of a large amount of DNA and labor-intensive procedures to achieve target DNA enrichment. The use of these methods for high-throughput and rapid TGS is challenging. The TAS methods are more sensitive and easier to scale up. Both methods involve a DNA sequencing step, and next-generation sequencing (NGS) is usually used for this [29,30,31]. Short-read NGS is commonly applied and has numerous advantages for TGS, including a high throughput, low cost per sample, and low error rate. Short-read NGS sequencing of amplicons has been widely used for MAS [17,32].

It is essential to build TGS systems for plants to make the entire TGS procedure more rapid, cheaper, and less labor-intensive for plant breeders worldwide. Short-read TAS requires expensive equipment and a high initial investment; therefore, the application of this approach in crop fields is not possible. Oxford Nanopore Technologies (ONT) provides a tiny sequencer, MinION, which allows sequencing to be run even in the field, as it only requires a USB connection to a laptop [33,34]. ONT sequencing has been used for TGS [35,36,37,38]. In our earlier study, we showed that we could quickly detect structural variations in promoters and coding regions, as well as new allele variants in large and complex plant genomes, by combining PCR amplification of individual genes and ligation-based barcoding [39]. However, when analyzing a large number of samples, this strategy can be costly and time-consuming because of the time required to amplify each individual gene and the length of time required to assemble the library.

In this study, we significantly improved and simplified the ONT-TAS procedure by using multiplex PCR and a rapid sequencing kit. We showed the efficiency of this novel strategy by the rapid sequencing of four genes (*Ahasl1*, *Ahasl2*, *Ahasl3*, *FAD2*) in forty sunflower (*Helianthus annuus*) plants and three genes (*Ppo*, *Wx*, *Lox*) in thirty wheat (*Triticum aestivum*) plants. The results revealed a high level of polymorphism scattered over the gene sequences. We also identified alleles of three sunflower genes (*Ahasl1*, *Ahasl3*, and *FAD2*) and two wheat genes (*Ppo-D1b* and *Lox1*) that differed by InDels and verified the results by PCR and Sanger sequencing. Overall, we demonstrated that the described ONT-TAS procedure is relatively simple, takes only a few days, and is not laborious for sequencing and genotyping target genes in sunflower (*Helianthus annuus*) and wheat (*Triticum aestivum*).

## 2. Materials and Methods

### 2.1. Plant Materials

For this study, sunflower seeds (Appendix A) obtained from V.S. Pustovoit All-Russian Research Institute of Oil Crops (Krasnodar, Russia) and spring bread wheat cultivars (Appendix A) of different geographical origins were used. Seeds were germinated at room temperature on wet filter paper disks. High-molecular-weight DNA was isolated from 200 to 500 mg of the material, which was homogenized in liquid nitrogen. DNA isolation was performed according to the published protocol (https://www.protocols.io/view/plant-dna-extraction-and-preparation-for389-ont-seque-bcvyiw7w, accessed on 4 September 2021). The concentration and quality of isolated DNA were assessed using a NanoDrop One UV-Vis Spectrophotometer (Thermo Scientific, Waltham, MA, USA). For amplification, equal concentrations of DNA were used, according to the manufacturer’s instructions.

### 2.2. Multiplex PCR

PCR amplification was conducted using specific primers (Appendix A), which were designed using Primer 3.0 software (https://www.bioinformatics.nl/cgi-bin/primer3plus/primer3plus.cgi, accessed on 1 September 2022). In multiplex PCR, two or more primer sets designed for the amplification of different targets are included in the same PCR reaction. Thus, the primer sets must be amplified under the same conditions. The PCR conditions were optimized for primer pairs, and optimal conditions for multiplex PCR were achieved.

For sunflower, PCR was performed in a 20 µL mixture containing 2× BioMaster LR HS-PCR (Biolabmix, Novosibirsk, Russia) reaction mixture, 0.2 µL of working solution of each primer, and 20 ng of DNA. Amplification was performed under the following temperature conditions using a mixture of four primer pairs: 35 cycles of denaturation at 94 °C for 20 s, primer annealing at 60 °C for 30 s, and elongation at 68 °C for 2 min.

For wheat, PCR was performed in a 20 µL mixture containing 2× BioMaster LR HS-PCR (Biolabmix, Novosibirsk, Russia) reaction mixture, 0.4 µL of working solution of each primer, 40 ng of DNA, and 5% DMSO. Amplification was performed under the following temperature conditions using a mixture of three primer pairs: 35 cycles of denaturation at 94 °C for 20 s, primer annealing at 65 °C for 30 s, and elongation at 68 °C for 3 min.

The multiplex PCR results were visualized via gel electrophoresis using a 1% agarose gel with ethidium bromide staining.

Multiplex amplicons were purified using 1× Agencourt AMPure XP Beads (Beckman Coulter, Pasadena, CA, USA) in accordance with the manufacturer’s instructions. The purified amplicon concentration and integrity were estimated using a NanoDrop One UV-Vis Spectrophotometer (Thermo Scientific, Waltham, MA, USA) and Qubit (Qubit dsDNA BR Assay Kits, Thermo Fisher Scientific, Waltham, MA, USA), respectively, and checked by gel electrophoresis. The PCR product concentrations were equalized for nanopore sequencing.

### 2.3. Library Preparation and Sequencing

For the native barcoding kit, we used phosphorylated primers for PCR amplification. For nanopore sequencing, a library was prepared from pooled samples using the nanopore native barcoding genomic DNA SQK-NBD110-24 (Oxford Nanopore Technologies, Oxford, UK), with some modifications in the process of using the NEBNext Companion Module for Oxford Nanopore Technologies Ligation Sequencing (New England Biolabs, MA, USA). Briefly, ~100 ng of each pooling sample in 4.5 µL was mixed with 0.5 µL Native Barcode and 5 µL Blunt/TA Ligase Master Mix and incubated on a Hula mixer for 10 min at room temperature. Purification by Agencourt AMPure XP Beads was performed during phosphorylation. Each of the 12 barcoded samples was resuspended in 2.7 µL of nuclease-free water and transferred to a new LoBind tube for adapter ligation. Then, ~32.5 µL was pooled, and barcoded amplicons were mixed with 10 µL NEBNext Quick Ligation Reaction Buffer (5X), 5 µL Quick T4 DNA Ligase, and 2.5 µL Adapter Mix II (AMII). Adapter Ligation Mix was incubated on a Hula mixer for 10 min at room temperature. Double washing was performed using 125 µL of Short Fragment Buffer (SFB). Incubation was performed in a water bath at 37 °C for 10 min and then for 5 min at room temperature. Sequencing was carried out using MinION and an SQK-LSK109 flow cell. Basecalling was performed by Guppy (Version 6.3.8).

For the rapid barcoding kit, each sample was mixed with 9 μL of the multiplexing PCR product (50 ng) and 1 μL of one rapid barcode. The mixture was incubated at 30 °C for 2 min, followed by incubation at 80 °C for 2 min. All barcoded DNA samples were pooled, and 800 μL of pooled DNA was mixed with an equal volume of AMPure XP Beads (Beckman Coulter, Pasadena, CA, USA). After 5 min of incubation at room temperature on a Hula mixer, the barcoded DNA was cleaned twice with 80% ethanol and eluted with 15 μL of Elution Buffer (EB). Incubation was performed at 37 °C for 30 min.

An aliquot of the barcoded DNA was used to obtain a total volume of 11 μL with EB. One microliter of Rapid Adapter F (RAP F) was added to the barcoded DNA, and the mixture was incubated at room temperature for 5 min. Then, 12 µL of barcoded amplicons was mixed with 37.5 µL of Sequencing Buffer II (SBII) and 25.5 µL of Loading Beads II (LBII). Sequencing was performed using MinION and an SQK-LSK109 flow cell. Basecalling was performed by Guppy (Version 6.3.8).

### 2.4. PCR Validation of InDels

To validate the InDels located in the sunflower and wheat genes, which were identified by amplicon sequencing, the primers listed in Appendix A were used. PCR was performed using Encyclo DNA polymerase (Evrogen, Moscow, Russia) according to the manufacturer’s instructions. The PCR conditions were specific for each InDel (Appendix A). To confirm the presence of InDels, the PCR products were directly sequenced by Sanger sequencing.

### 2.5. SNP Calling

For variant calling, the ONT reads were mapped to the reference sequences using minimap2 [40]. The obtained BAM files were sorted and indexed using Samtools [41].

Primary SNP calling was carried out using Calir3 (https://github.com/HKU-BAL/Clair3, accessed on 28 July 2024) with the r941_prom_hac_g360+g42 model. The obtained vcf files of different samples were filtered (--remove-filtered-all--minQ 10) and merged using vcftools [42]. The phylogenetic tree was built from the merged vcf files using VCF2PopTree [43]. The SNPs were annotated using SNPeff [44].

## 3. Results

### 3.1. Three Approaches for Nanopore Amplicon Sequencing

Three amplicon sequencing experiments were performed to determine the optimal library preparation method for sunflower and wheat collections (Table 1).

For sunflower, we choose three genes (*Ahasl1*, *Ahasl2*, and *Ahasl3*) involved in herbicide resistance [21,22,23,24] and one gene (*FAD2*) that is responsible for the high oleic acid content of sunflower seeds [26]. The first ONT sequencing experiment was performed using the native barcoding kit. Primers (Figure 1A) were designed to amplify the four sunflower genes (*Ahasl1*, *Ahasl2*, *Ahasl3*, and *FAD2*) (Figure 1C). The PCR products were barcoded according to the sunflower genotype using a native barcoding kit and sequenced on a single MinION flow cell. The sequencing procedure took 2 h and resulted in 341,477 reads. The number of reads per gene per sample varied between 6222 and 20,850, providing sufficient gene coverage for downstream analysis. Each primer pair was amplified separately for the sunflower genotypes. All PCR products were purified, pooled according to the sample number, and barcoded according to the manufacturer’s protocol. Owing to the limited number of barcodes, 12 barcodes were first loaded into the flow cell, and another part of the barcodes was loaded after washing. An analysis of allelic variants was carried out by comparing the obtained reads with the reference sequences for *Ahasl1* (NC_035441.2), *Ahasl2* (NC_035438.2), *Ahasl3* (AY541458.1), and *FAD2* (FJ791046.1).

To make the ONT-TAS procedure more robust and routine, we performed two important modifications in the next experiment. First, we optimized the multiplex amplification procedure and achieved amplification of all four genes in a single tube during one PCR reaction (Figure 1D). Amplicons were successfully generated from the 40 sunflower genotypes. Each amplicon contained the following PCR products: *Ahasl1* (2000 bp), *Ahasl2* (2010 bp), *Ahasl3* (1921 bp), and *FAD2* (2000 bp). Second, we used a rapid barcoding kit instead of the native barcoding kit. The rapid barcoding kit performed transpose-mediated barcoding and allowed the barcoding procedure to be completed within 10 min. These changes in the ONT-TAS significantly reduced the time required for library preparation and made it less laborious. Thus, 40 sunflower samples were amplified using a mixture of the four primer pairs, resulting in 40 amplicons at a time with four different products of each size. A mapping of the reads to the reference genes showed that all target genes were sequenced for all sunflower genotypes, implying the successful amplification of the genes by multiplex PCR. We evaluated the length of the obtained reads and the target gene coverage resulting from the application of the native barcoding kit (Figure 1F) and the rapid barcoding kit (Figure 1G). We found that the median coverage values for *Ahasl1*, *Ahasl2*, *Ahasl3*, and *FAD2* obtained using the rapid barcoding kit varied within the ranges of 13×–214×, 198×–575×, 360×–1097×, and 14×–240×, respectively (Figure 1H). These values were lower than those for the native barcoding kit. However, the obtained coverage values for the rapid barcoding kit were still sufficient for SNP calling and downstream analyses.

Previously, we used a native barcoding kit to sequence target wheat genes [39]. In this study, we applied the multiplex + rapid barcoding strategy to sequence three wheat genes (*Ppo*, *Wx*, and *Lox*) of 30 wheat lines. To amplify these target genes, primers were designed for the coding and promoter regions specific to the three subgenomes (Figure 1B). Although the primers were designed for multiplex PCR, the PCR conditions for each primer pair were optimized to achieve optimal multiplex conditions. Subsequently, 30 amplicons containing three different-sized products, *Ppo* (2200 bp), *Wx* (2900 bp), and *Lox* (3000 bp), were obtained (Figure 1E). The obtained reads were aligned to the reference sequences of wheat genes from NCBI: EF070148.1 (*Ppo-A1b*), GQ303713.1 (*Ppo-B1a*), EF070150.1(*Ppo-D1b*), GQ166692.1 (*Lox1*), HQ406780.1 (*Lox-B1b*), KC679302.1 (*Lpx-D1*), MT048401.1 (*Wx-A1q*), KF861808.1 (*Wx-B1l*), and LC373576.1 (*Wx-D1g*). The sequencing yielded >200,000 reads, resulting in up to 40× target gene coverage.

Thus, the results showed that TAS using a combination of multiplex PCR and a rapid barcoding kit allowed for rapid sequencing of target genes with high coverage values.

### 3.2. SNP Validation in Sunflower Varieties

We then performed SNP calling using Clair3 [45], followed by variant annotation using SNPeff [46]. The resulting vcf files contained 246 SNPs among the target genes of the sunflower varieties. *Ahasl1* (Figure 2A) had the highest number (189 SNPs), of which 72% (137 SNPs) belonged to synonymous SNPs, and 28% (52 SNPs) were non-synonymous SNPs (nsSNPs). *Ahasl3* (Figure 2B) showed 6 nsSNPs and 48 synonymous SNPs, representing 22% (54 SNPs) of all identified SNPs. Overall, according to the SNP annotation, 23.6% (58 SNPs) of the identified SNPs were missense, while more than 76% (188 SNPs) were synonymous.

In terms of the number of SNPs, the most divergent sunflower sample was RHA450, which had 43 SNPs, including 16 synonymous and 6 non-synonymous SNPs for *Ahasl1* and 17 synonymous SNPs and 4 nsSNPs for Ahasl3. The minimum number of SNPs was observed for *Ahasl1* (one synonymous SNP) and *FAD2* (two synonymous SNPs) in the VK464 and ZS samples, respectively.

Further, the proportions of 12 mutation types were considered (Figure 2C). The most frequent mutations were C-to-T (34 SNPs), A-to-G (30 SNPs), A-to-C (30 SNPs), G-to-A (29 SNPs), G-to-T (25 SNPs), and G-to-C (19 SNPs). Only the G-to-T mutation was 60% represented by non-synonymous SNPs, while the rest (A-to-G, G-to-A, and C-to-T) were 70% or more represented by synonymous mutations. Among the 12 possible substitutions, only 4 (A-to-C, G-to-C, A-to-T, and T-to-A) were represented by fully synonymous mutations of all substitutions, of which the least represented substitutions were A-to-T and T-to-A, with one and two synonymous SNPs, respectively. It was found that those synonymous mutations that changed the GC content represented about 35% of all mutations and were represented by A-to-C (40%), A-to-G (40%), T-to-C (16%), and T-to-G (4%), which were not synonymous with A-to-G (eight SNPs), T-to-C (six SNPs), and T-to-G (two SNPs).

Based on the analysis of four target genes in the sunflower collection, *Ahasl1* (Figure 2C) accounted for the largest number of variations (189 SNPs), most of which (137 SNPs) were synonymous. The remaining 52 SNPs were non-synonymous substitutions. The *Ahasl3* gene (six SNPs) had almost nine times fewer non-synonymous substitutions than the *Ahasl1* gene in the sunflower collection. The smallest number of SNPs was found in *Ahasl2* (one SNP) and *FAD2* (two SNPs). To further explore the genetic diversity of the sunflower collections using ONT-TAS data, we constructed a phylogenetic tree based on the detected SNPs. This analysis revealed five groups of sunflower genotypes (Appendix A) that differed significantly by the SNPs in *Ahasl1*. Additionally, we identified 1, 3, 1, 1, and 34 sunflower genotypes that clustered into five groups according to their *Ahasl3* SNPs (Appendix A). The results showed significant diversity of the *Ahasl1* and *Ahasl3* genes in the sunflower collection, unraveled by the ONT-TAS procedure.

Using a similar approach, we analyzed the diversity of the target wheat genes. We detected two, three, and three different alleles for the *Ppo-A*, *Ppo-B*, and *Ppo-D* genes, respectively. Six, one, and two alleles were identified for the *Wx-A*, *Wx-B*, and *Wx-D* genes, respectively. Additionally, one, six, and one allele were found for the *Lox-A*, *Lox-B*, and *Lox-D* genes, respectively. Thus, the *Wx-A* and *Lox-B* genes had the highest number of alleles in our collections.

Taken together, the ONT-TAS approach allowed us to rapidly characterize the genetic diversity of the target genes in sunflower and wheat in the germplasm collections.

### 3.3. Analysis of Structural Variations

Structural variations were discovered during the analysis of the sunflower SNPs. In the sunflower genotypes analyzed, a deletion of approximately 10 bp was found within the tandem regions of *FAD2* (Appendix A). This deletion, which occurs in repetitive regions, highlights the complexity of assembling and analyzing long tandem repeats in these plants. Additionally, a deletion of approximately 10 bp was detected in *Ahasl1*. This deletion was observed in ten sunflower samples and was located in the coding region of the gene. Upon comparing the obtained *Ahasl3* reads with the reference sequences, an insertion of approximately 15 bp was discovered (Figure 3A). This insertion was located within the coding region of the gene in RHA450 and ZS samples. Flanking primers were designed for all identified InDels for PCR analysis (Figure 3C) and Sanger sequencing (Figure 3D).

Structural variations were identified during the analysis of the wheat ONT amplicon sequencing data. InDels within the *Ppo-D1* gene (~30 bp) were identified in a single sample, whereas InDels in the *Lox1* gene (~300 bp) were identified in four samples (Figure 2B). Both InDels were confirmed by Sanger sequencing (Figure 3D).

### 3.4. Economic Aspects of ONT-TAS

The estimated price of sequencing a single sample using this approach was approximately USD 16.15, or USD 4 per gene (Table 2). The application of multiplex PCR together with the rapid barcoding kit further reduced the cost of ONT-TAS sequencing to USD 13.61, or USD 3.4 per gene. This price was calculated considering that up to six sequencing runs could be carried out for a single MinION flow cell. This price is even lower than that of Sanger sequencing (for example, USD 5.25 per sample proposed by https://eurofinsgenomics.com/en/products/pricing/, accessed on 24 September 2024). However, in contrast to Sanger sequencing, the ONT-TAS procedure can easily be used for long PCR products (up to 3Kb in this study).

## 4. Discussion

In this study, we showed that the combination of multiplex PCR and a rapid barcoding procedure provides a rapid and cost-effective approach to detect SNPs and SVs in full-length target genes across multiple samples of plant collections. We verified this approach to determine the genetic variation in four (*Ahasl1*, *Ahasl2*, *Ahasl3*, and *FAD2*) sunflower genes and three (*Ppo*, *Wx*, and *Lox*) wheat genes in collections comprising 30 and 40 samples, respectively. The obtained data explicitly showed the distribution of SNPs and InDels along full-length genes, providing a complete picture of the target gene diversity in the germplasm collection.

### 4.1. Multiplex PCR and Rapid Barcoding Kit Make Gene Sequencing Procedure Easier and Faster

Previously, we [39] and others [47] used native barcoding kits for amplicon sequencing. We performed a comparative evaluation of the native barcoding kit and the rapid barcoding kit + multiplex. These approaches differ in terms of the price and time required. For the native barcoding approach, the amplicons of each wheat sample (92 amplicons in our case) were individually amplified, purified, and pooled to obtain the final samples for barcoding. The phosphorylated PCR amplicons from each sample were barcoded, combined in one tube, and sequenced. Although this approach resulted in longer and intact reads, it was more expensive because it required a third-party reagent (Ligase Master Mix) and more PCR reactions. Additionally, the library preparation for the native barcoding kit was more time-consuming and took nearly 3 h, compared to the 1.5 h required for the rapid barcoding kit (Figure 4).

While native barcoding is more time-consuming and incurs a slightly higher cost per sample, it produces significantly longer reads compared to the rapid barcoding kit. However, the increased coverage provided by the rapid kit may yield a sufficient number of reads corresponding to full-length amplicons. Longer reads can enhance the accuracy and reliability of variant detection [48]. Additionally, the Ligation Kit supports more complex library preparation, including size selection and control over the read length, which can further improve the SNP calling accuracy by minimizing amplification bias. Consequently, although both kits can be effectively used for SNP calling, the choice between them may depend on the specific requirements of the sequencing project, such as the trade-off between speed and accuracy, as well as the desired depth of information.

### 4.2. Advantages of ONT-TAS

ONT amplicon sequencing has been widely used for 16S ribosomal RNA (rRNA) sequencing in metagenomic studies. This method offers significant advantages over traditional short-read sequencing techniques, allowing for comprehensive analyses of complex microbial communities [49,50]. There are also several reports on the application of ONT-TAS for sequencing eukaryotic protein-coding genes [37,38,39,47,50,51]. These and our current studies highlight the key advantages of ONT-TAS, which make it useful for broader applications for plant genotyping and marker-assisted selection (MAS). First, ONT-TAS allows for the accurate identification of SNPs across multiple samples simultaneously. Second, ONT-TAS can be used in combination with multiplex PCR to reduce the price and turnaround time. Third, in addition to SNPs, ONT-TAS can also characterize structural variants owing to its long-read capabilities. Fourth, the real-time sequencing capability of nanopore technology allows for an immediate data analysis and interpretation of results. Finally, the portable nature of nanopore sequencers enables their use in field settings, which makes them suitable for rapid genotyping in diverse environments. These features offer several distinct benefits for the application of ONT-TAS in other areas including CRISPR validation. ONT-TAS allows for comprehensive coverage of target regions and adjacent sequences and helps to accurately identify structural variants, confirming the integrity of the modified genomic context, which is often missed by short-read technologies. This is especially important when several sgRNAs are applied, and different structural modifications can occur [52,53,54].

The ONT-TAS approach could become an effective tool used in the pre-breeding stage to identify novel and desirable allelic gene variants in large plant collections possessing hundreds to thousands of genotypes. Useful variants could then be traced during subsequent crosses using ONT-TAS. Alternatively, the obtained sequences of novel variants could be used to design SNP-specific high-throughput markers such as KASP [15]. Such an approach, which combines full-length gene sequences and SNP-specific markers, will lead to the discovery of new alleles and more precise breeding when target genes are known.

### 4.3. Current Limitations of ONT-TAS

There are still some obstacles that limit the broad application of ONT-TAS in full-length gene sequencing. First, there is no user-friendly software for easy downstream analysis of ONT-TAS data, including variant calling, phylogenetic analysis, and novel and known allele identification. Several modern user-friendly software programs have been developed for pathogen detection (e.g., SARS-CoV-2), including ONTdeCIPHER [55], MinoTour [56], and ONT-DART [57]. These programs allow users to trace sequencing progress in real time, preprocess data, identify variants, and perform phylogenetic analyses using nanopore amplicon data. The adaptation of these programs to ONT-TAS data will further accelerate the utility of full-length gene sequencing in breeding programs; however, it will require a comprehensive database of known alleles for the target genes.

Secondly, the multiplex PCR procedure needs to be optimized for individual primer combinations, which can be time-consuming and challenging. In this work, we followed these steps to determine the best conditions for multiplex PCR optimization: (1) design primers with the same annealing temperature; (2) optimize the melting temperature (Tm) separately for each primer combination using gradient PCR; (3) use the Tm that results in minimal non-specific amplification for all primer pairs; (4) employ high-precision DNA polymerases (e.g., Pfu) to improve the accuracy of the reaction; (5) add DMSO to achieve a final concentration of up to 5%.

Thirdly, ONT reads possess a relatively high error rate compared to traditional short-read sequencing methods, such as Illumina [48]. Although single-nucleotide sequencing errors can be easily discriminated as they occur randomly along the reads, homopolymeric regions are particularly problematic. Errors in these areas often manifest as deletions, making it difficult to accurately determine the length of homopolymers [48]. The error rate in the ONT data is directly related to the version of the library preparation chemistry and the flow cell type. In this study, we used V10 chemistry and R9 nanopore flow cells, which are prone to producing reads with high error rates. However, a combination of the R10 flow cell (for example, R10.4.1) and the latest Q20+ ONT chemistry can improve ONT-TAS data [58,59]. Furthermore, ‘duplex base calling’ has been introduced, which can provide Q30 (99.9% accuracy) single-molecule reads, achieving an error rate comparable to that of Illumina sequencing. Recent studies have compared the performance of R9 and R10 nanopore sequencing chemistries, focusing on their read accuracy and assembly capabilities [60,61]. Trial experiments demonstrated that the R10 chemistry, particularly in its R10.4 iteration, provided a very low error rate and improved the consensus accuracy compared to R9.4.1 [60,61]. The new ONT chemistry and flow cell will further facilitate the routine use of ONT-TAS for marker-assisted selection, wherein hundreds of individuals are typically genotyped within a limited timeframe.

## 5. Conclusions

This study demonstrated the potential of the ONT-TAS method for high-throughput genotyping of full-length target genes in plant collections. The combination of multiplex PCR and a rapid sequencing kit for the ONT-TAS approach provides sufficient information for SNP calling and InDel detection in target genes across multiple genotypes and is useful for the rapid identification of known and novel alleles of desired genes.

## Figures and Tables

**Figure 1 biology-14-00138-f001:**
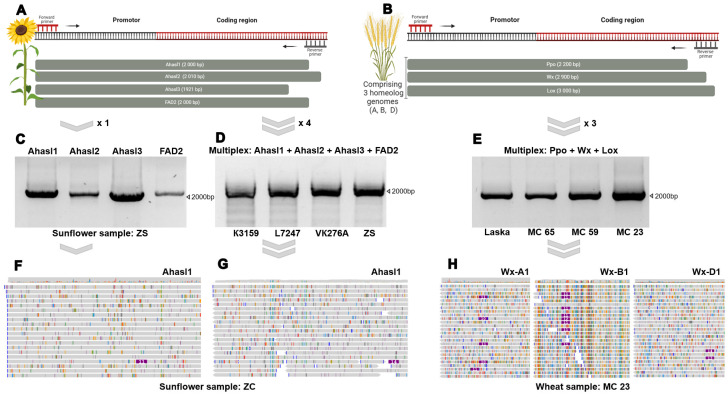
Schematic depicting the primer positions for the four sunflower genes (**A**) and three wheat genes specific to subgenomes A, B, and D (**B**). Sunflower primers were used in two experiments: (**C**) PCR products were amplified separately, pooled, barcoded, and sequenced using the native barcoding kit; the PCR included four primer pairs for sunflower (**D**) and three primer pairs for wheat (**E**), and the multiplex products were sequenced using the rapid barcoding kit. Read alignment to the reference sequence of the *Ahasl1* gene after using the native barcoding kit (**F**), the rapid barcoding kit (**G**), and the sequence of three *Wx* genes (A, B, and D subgenomes) (**H**).

**Figure 2 biology-14-00138-f002:**
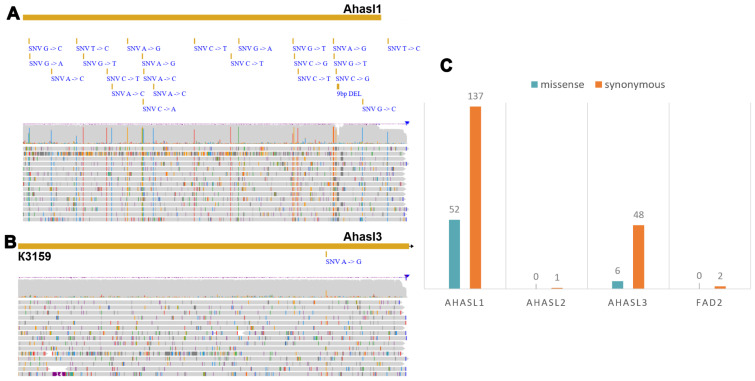
A sequence analysis of *AHASL* genes helps match the reference sequence with the obtained reads for *Ahasl1* (**A**) and *Ahasl3* (**B**) for SNP identification. Two long yellow lines indicate reference genes and short bars below indicate SNPs. The bottom plots display the ONT read alignments to the genes, with colors indicating different SNPs. (**C**) A histogram illustrating the counts of synonymous (orange) and non-synonymous (blue) SNPs detected in the four sunflower genes. The yellow line indicates the reference gene, with short bars underneath indicating SNPs.

**Figure 3 biology-14-00138-f003:**
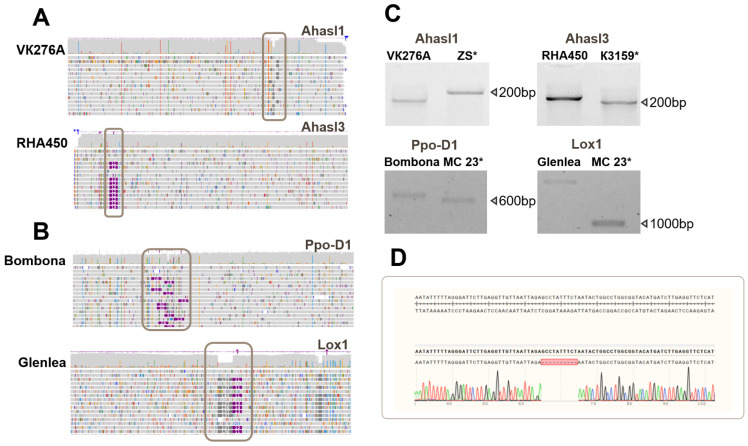
Sequence analysis of target genes and verification of insertions and deletions (InDels). ONT-TAS read alignment to the reference sequences of sunflower (**A**) and wheat (**B**) genes. InDels are highlighted by gray rectangles, with gray indicating deletions and purple indicating insertions. (**C**) PCR results using specific primer pairs flanking the InDels for sunflower genes (*Ahasl1* and *Ahasl3*) and wheat genes (*Ppo-D1* and *Lox1*). Reference alleles are indicated by asterisks (*). (**D**) An example of a chromatogram obtained by Sanger sequencing of amplicons with InDels. A deletion is indicated by red block.

**Figure 4 biology-14-00138-f004:**
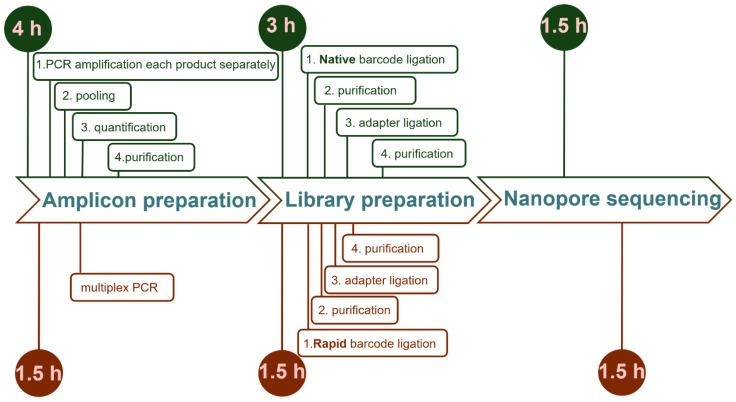
Comparison of the timelines of the two amplicon sequencing approaches. The top line (green) shows amplicon sequencing using a native barcoding kit and single PCR. The bottom line (red) shows amplicon preparation using the rapid barcoding kit and multiplex PCR.

**Table 1 biology-14-00138-t001:** Summary of nanopore sequencing experiments.

Experiment	Organism	Number of Samples	Multiplex	Number of Genes	Library Preparation	Number of Reads per Sample
Experiment 1	Sunflower	23	no	4	Native barcoding kit	>6000
Experiment 2	Sunflower	40	yes	4	Rapid barcoding kit	>2000
Experiment 3	Wheat	30	yes	3	Rapid barcoding kit	>2000

**Table 2 biology-14-00138-t002:** The estimated prices for sequencing four genes for a single sample using the native barcoding kit and rapid barcoding kit.

	Native Barcoding Kit, Cost/Sample USD	Rapid Barcoding Kit and Multiplex PCR for 4 Genes, Cost/Sample USD
DNA extraction	1.77	1.77
Amplification and purification	2.72	0.68
Barcoding and library preparation	8.3 (Native barcoding kit 96 V14)	10.3 (Rapid barcoding kit 96 V14)
Third-party consumables for library preparation kit	2.5 (Ligation Mix)	0
Flow cell price (up to 6 individual runs)	0.86	0.86
Total cost	16.15	13.61

## Data Availability

The nanopore data produced for this study are available in the Sequence Read Archive (SRA) NCBI under the Bioproject Accession PRJNA1213285.

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
