# Peer review of "Cost-Effective Detection of SNPs and Structural Variations in Full-Length Genes of Wheat and Sunflower Using Multiplex PCR and Rapid Nanopore Kit"

_biology, 2025, doi:10.3390/biology14020138_

Round 1

Reviewer 1 Report

Comments and Suggestions for Authors

Comments to the Authors

The manuscript entitled “Cost-effective Detection of SNPs and Structural Variations in Full-length Genes of Wheat and Sunflower Using Multiplex PCR and Rapid Nanopore Kit” by Polkhovaskaya et al. is interesting and it provides a strong base for genomics-assisted breeding approach targeting multiple genes. The manuscript content is timely with appropriate tables and figures and insightful discussion. However, the following minor concerns need to be addressed before proceeding further.  

In lines 22-23, the gene names should be italicized and the same has to be reviewed thoroughly throughout the manuscript.

Lines 23-25 should be revised with more details and better conveyance of the findings. 

In Figure 1 panels F, G, and H are not clear. These should be revised with better quality.

In continuation to the previous comment, the quality of Figure 2 also needs to be enhanced with clearly visible text.

In lines 357-366, the cost and time discussed in the discussion section appear to be the main part of the results section. The authors can think of moving this content to the results section appropriately.

Also, suggested revising the discussion section with an appropriate sub-heading with a clear emphasis on the comparison of the findings with earlier studies.     

Author Response

Q1: In lines 22-23, the gene names should be italicized and the same has to be reviewed thoroughly throughout the manuscript.

A1: The change has been made in the text of the MS.

Q2: Lines 23-25 should be revised with more details and better conveyance of the findings. 

A2. The abstract was modified.

Q3: In Figure 1 panels F, G, and H are not clear. These should be revised with better quality.

In continuation to the previous comment, the quality of Figure 2 also needs to be enhanced with clearly visible text.

A3: The Figures have been modified.

Q4: In lines 357-366, the cost and time discussed in the discussion section appear to be the main part of the results section. The authors can think of moving this content to the results section appropriately.

A4: The change has been made in the text of the MS.

 Q5: Also, suggested revising the discussion section with an appropriate sub-heading with a clear emphasis on the comparison of the findings with earlier studies.     

A5: Discussion section has been revised 

Reviewer 2 Report

Comments and Suggestions for Authors

In this manuscript, the authors used targeted amplicon sequencing with Oxford Nanopore Technologies (ONT-TAS) to rapidly sequence full-length genes and identify allelic variants in sunflower and wheat collectionsThis work was well designed. The methods of data collection and its presentation are appropriate.

Comments:

--- In the abstract, line 29: not pro-grams. There are several typos in the manuscript. Please thoroughly review the manuscript and correct it.

---  Please format all scientific names of species in italics. For instance, on page 3, lines 100-101. Please thoroughly review the manuscript and correct them

Author Response

Q1: In the abstract, line 29: not pro-grams. There are several typos in the manuscript. Please thoroughly review the manuscript and correct it.

A1: We carefully checked the MS and make some corrections accordingly.  

 Q2: Please format all scientific names of species in italics. For instance, on page 3, lines 100-101. Please thoroughly review the manuscript and correct them

A2: The change has been made in the text of the MS.

Reviewer 3 Report

Comments and Suggestions for Authors

The study introduces a highly relevant and cost-effective method for detecting SNPs and structural variations (SVs) in target genes of wheat and sunflower using Oxford Nanopore Technologies (ONT) sequencing. The incorporation of multiplex PCR and the rapid sequencing kit is a valuable innovation, as it significantly reduces the cost and time required for genotyping large numbers of plant samples. The focus on SNPs and InDels (insertions/deletions) in genes that are relevant to plant breeding (e.g., herbicide resistance, oil content) adds to the practical importance of this study. The targeted genes (Ahasl1, Ahasl2, Ahasl3, FAD2 in sunflower, and Ppo, Wx, Lox in wheat) are integral to breeding programs, making this technique highly applicable in agricultural biotechnology. The integration of ONT sequencing with barcoding kits is also a notable strength, especially for high-throughput applications. The results are validated through Sanger sequencing, which strengthens the reliability of the findings. SNP and structural variant identification is clearly explained, and phylogenetic analysis of the SNPs further supports the robustness of the method for identifying genetic diversity. The manuscript presents detailed data, including SNP calling, variant annotation, and structural variation detection, which provides a comprehensive view of the genetic variation in the studied crops.

However, there are some problems, which must be solved before it is considered for publication. If the following problems are well-addressed, I believe that the paper can be published.

(1) A significant drawback of ONT sequencing is its relatively high error rate, especially in homopolymeric regions. While the authors acknowledge this limitation, the study could benefit from more detailed discussion on strategies for minimizing these errors. For instance, the authors mention the potential improvement with R10.4.1 flow cells and Q20+ ONT chemistry; however, this improvement is not sufficiently explored. So please provide data from trials using these updated technologies to strengthen their claims.

(2) Although the study compares the native barcoding kit with the rapid barcoding kit, there is no in-depth analysis of the strengths and weaknesses of each method beyond sequencing costs and time. I believe that a more detailed comparison of read accuracy, depth of coverage, and the impact of these factors on downstream analyses such as SNP calling would offer a clearer picture of the trade-offs between these methods.

(3) The study involves 40 sunflowers and 30 wheat genotypes, which, although a reasonable number, could be expanded to provide more robust conclusions, particularly for structural variation and rare alleles. A larger, more diverse set of samples would increase the generalizability of the findings.

(4) The authors note that there is no user-friendly software for downstream analysis of ONT-TAS data. While this is an important limitation, the paper does not suggest any potential solutions or collaborations with bioinformatics tools that could facilitate data analysis for broader adoption of this approach in plant breeding.

(5) The data produced from this study are mentioned as being available in the Sequence Read Archive (SRA), but it would be helpful to provide the exact accession numbers for readers to easily access and verify the data. So please provide it if you can.

(6) While the multiplex PCR results are described in detail, a more in-depth discussion on the optimization and reproducibility of the PCR across different plant varieties (including potential issues with primer mismatches or gene amplifications in certain genotypes) would improve the robustness of the method. So I believe that the depth discussion is necessary

(7) I hold that the estimated cost comparison between the native barcoding kit and the rapid barcoding kit is helpful, but the paper could expand on the practical implications of these cost reductions. For example, the potential for large-scale applications in plant breeding and the associated cost savings should be discussed in more detail.

(8)While the study is focused on wheat and sunflower, validating the method in additional crop species would be valuable. This would demonstrate the versatility of the method and its broader applicability in plant breeding programs. So, please provide further validation across different crops.

(9) The discussion section could be strengthened by elaborating on the specific applications of this technique in plant breeding, particularly how this method can be integrated into existing workflows for marker-assisted selection (MAS). So please enhance discussion on applications.

(10)Expanding on the strategies for error correction in ONT sequencing, such as the potential for using consensus sequencing methods or hybrid approaches with other platforms like Illumina, would make the paper more comprehensive. So, please explore error mitigation strategies.

Author Response

Q1: A significant drawback of ONT sequencing is its relatively high error rate, especially in homopolymeric regions. While the authors acknowledge this limitation, the study could benefit from more detailed discussion on strategies for minimizing these errors. For instance, the authors mention the potential improvement with R10.4.1 flow cells and Q20+ ONT chemistry; however, this improvement is not sufficiently explored. So please provide data from trials using these updated technologies to strengthen their claims.

А1: We added the corresponding information to the end of Discussion section of the MS.

Q2: Although the study compares the native barcoding kit with the rapid barcoding kit, there is no in-depth analysis of the strengths and weaknesses of each method beyond sequencing costs and time. I believe that a more detailed comparison of read accuracy, depth of coverage, and the impact of these factors on downstream analyses such as SNP calling would offer a clearer picture of the trade-offs between these methods.   

A2: The main difference between these two methods, in terms of data, is the read length, as described in the results. The Rapid Barcoding Kit provides a much smaller N50 and lower sequencing coverage. This trade-off comes at the expense of reduced cost and time. We agree that a more detailed comparison of the Rapid and Ligation Kits is necessary, and we have added this information to the end of Section 4.1 of the discussion. 

Q3: The study involves 40 sunflowers and 30 wheat genotypes, which, although a reasonable number, could be expanded to provide more robust conclusions, particularly for structural variation and rare alleles. A larger, more diverse set of samples would increase the generalizability of the findings.

А3. We appreciate the reviewer’s suggestion to include additional samples in our analysis. However, for this study, we focused on two species that significantly differ in genome size (3 Gb for sunflower vs. 16 Gb for wheat) and phylogeny (monocotyledon and dicotyledon). Furthermore, sunflower is known to be a challenging species for PCR and sequencing. Therefore, we believe that the successful application of ONT-TAS to these highly diverse species is sufficient to support our stated conclusions.

Additionally, the methods used in our study (multiplex primer design, PCR, library preparation, and ONT sequencing) are well established, and their optimization for different species is an ongoing endeavor. Thus, we contend that the current scope of our study adequately supports our conclusions, and adding more species and samples at this stage would not significantly enhance the strength of our findings. 

Q4: The authors note that there is no user-friendly software for downstream analysis of ONT-TAS data. While this is an important limitation, the paper does not suggest any potential solutions or collaborations with bioinformatics tools that could facilitate data analysis for broader adoption of this approach in plant breeding.

А4: The information has been added to the Discussion section of the MS.

Q5: The data produced from this study are mentioned as being available in the Sequence Read Archive (SRA), but it would be helpful to provide the exact accession numbers for readers to easily access and verify the data. So please provide it if you can.

A5: Data related to crops and varieties was uploaded to NCBI under Bioproject number PRJNA1213285.

Q6: While the multiplex PCR results are described in detail, a more in-depth discussion on the optimization and reproducibility of the PCR across different plant varieties (including potential issues with primer mismatches or gene amplifications in certain genotypes) would improve the robustness of the method. So I believe that the depth discussion is necessary

A6: We have added the information about multiplex optimization in the Discussion section.

Q7: I hold that the estimated cost comparison between the native barcoding kit and the rapid barcoding kit is helpful, but the paper could expand on the practical implications of these cost reductions. For example, the potential for large-scale applications in plant breeding and the associated cost savings should be discussed in more detail.

A7: We agree with this comment. We added a paragraph about possible application of ONT-TAS during pre-breeding sarge when a number of individual genotypes need to be analyzed.

Q8: While the study is focused on wheat and sunflower, validating the method in additional crop species would be valuable. This would demonstrate the versatility of the method and its broader applicability in plant breeding programs. So, please provide further validation across different crops.

А8: Please, check the answer above (Q3). 

Q9: The discussion section could be strengthened by elaborating on the specific applications of this technique in plant breeding, particularly how this method can be integrated into existing workflows for marker-assisted selection (MAS). So please enhance discussion on applications.

A9: We agree with this comment. We added a paragraph about possible application of ONT-TAS during pre-breeding sarge when a number of individual genotypes need to be analyzed.

Q10: Expanding on the strategies for error correction in ONT sequencing, such as the potential for using consensus sequencing methods or hybrid approaches with other platforms like Illumina, would make the paper more comprehensive. So, please explore error mitigation strategies.

A10: We agree that error correction is an important subject. However, we would not like to explore different strategies for error correction, particularly the utilization of Illumina, as it complicates ONT-TAS and is not necessarily required. We believe that discussions on this and other error correction strategies are more relevant to de novo assembly approaches and genome sequencing papers. Nevertheless, we have added more information about the importance of utilizing modern ONT chemistry to significantly reduce the error rate.